# Research on Laser-Induced Damage Post-Restoration Morphology of Fused Silica and Optimization of Patterned CO_2_ Laser Repair Strategy

**DOI:** 10.3390/mi14071359

**Published:** 2023-06-30

**Authors:** Xiao Shen, Ci Song, Feng Shi, Ye Tian, Guipeng Tie, Shuo Qiao, Xing Peng, Wanli Zhang, Zhanqiang Hou

**Affiliations:** 1College of Intelligence Science and Technology, National University of Defense Technology, Changsha 410073, Chinapeng_xing22@163.com (X.P.); zhangwanli17@nudt.edu.cn (W.Z.);; 2Hunan Key Laboratory of Ultra-Precision Machining Technology, National University of Defense Technology, Changsha 410073, China; 3Laboratory of Science and Technology on Integrated Logistics Support, National University of Defense Technology, Changsha 410073, China

**Keywords:** laser-induced damage restoration, CO_2_ laser, patterned restoration strategy

## Abstract

Fused silica has become the preferred optical material in the field of inertial confinement fusion (ICF) due to its excellent performance; however, these costly optical elements are vulnerable, and their manufacture is time-consuming. Therefore, the restoration of laser-induced damage for these optical elements is of great value. To restrain the post-restoration raised rim problem in the CO_2_ laser repair process to improve the restoration quality, the separate influences of key parameters of laser power, irradiation duration, and laser beam diameter on post-restoration pit morphology are compared in combined simulation and experimental studies. An optimized, patterned CO2 laser strategy is proposed and verified; the results indicate that, with the strategy, the rim height decreases from 2.6 μm to 1.52 μm, and maximal photo thermal absorption is decreased from 784.2 PPM to 209.43 PPM.

## 1. Introduction

As we are faced with problems of traditional non-renewable energy resource depletion and serious environmental pollution, nuclear energy has attracted a lot of attention as a clean and efficient energy. In recent years, due to its small scale, high cooling efficiency, easy maintenance, and avoidance of nuclear leakage and the nuclear waste disposal problem, intense laser-based inertial confinement fusion (ICF) has become a hot research topic [1]. In ICF, multiple high-throughput lasers are simutaneously transmitted through the laser driver first, a huge amount of energy will be then generated as the result of target sphere implosive compression by high power laser irradiation. Currently, the main ICF devices under construction are the one housed at the National Ignition Facility (NIF) in the United States [2], the Laser Mega Joule (LMJ) in France [3,4] and China’s Shenguang III laser device [5,6].

The optimal component in ICF is excellent resistance to laser damage; in NIF, the total energy used for laser ignition is 1.8 megajoules, which leads to the damage threshold demand for the key lens material being over 14 J/cm^2^ (351 nm, 3 ns). In this case, fused silica has become the preferred material for large aperture components, due to its high theoretical intrinsic damage threshold and excellent permeability over the visible, ultraviolet, and infrared spectra. The number of large aperture optic elements (diameter over 430 mm) which are made of fused silica in NIF is more than 2700, and the number is over 2400 for LMJ [7]. However, researchers also found that fused silica components produced via cold processing are seriously damaged, and they are irradiated by a laser flux with a power total that is well below the theoretical intrinsic damage threshold. Resulting from surface and subsurface defects generated during fabrication, the actual damage threshold of a component’s surface is from 8 to 10 J/cm^2^ only [7]; as a consequence, in ICF systems, surface laser-induced damage is inevitable and, worse still, the dimensions of these laser-induced defects grow rapidly once it has been damaged [8,9]. The damage seriously threatens the lifespan of components. In accordance with NIF standards, when laser damage area (including repaired area) takes over 3 percent of whole light-passing region, the component reaches the end of its life and must be replaced [10]. Considering the manufacturing and maintenance cost, as well as the stability of fused silica optical components in high-power laser systems, research concerning improving the laser-induced damage threshold and the suppressing laser damage growth is of great value.

To restrain laser damage growth under subsequent laser irradiation, and prolong the lifespan of the components to maintain long-term stable operation and reduce the cost of the laser device, it is of great necessity to restore the surface. As it is one of the most effective and widely applied methods, since 2001, many researchers have conducted research on the field of CO_2_ laser repair techniques. Kubota A used a CO_2_ laser to repair surface damage craters with dimensions ranging from 4 to 30 μm [11]. Adams restored a damage crater with a diameter of over 200 μm using an optimized repair scheme [12]. Based on the spiral scanning method, Bass [13] and Guss [14] applied a galvanometer to a CO_2_ repair process, and the results indicate that, after restoration, 7 out of 10 repair points are capable of bearing 400 laser shots with a laser flux from 20 to 25 J/cm^2^. Zhou found that a conical pit with an angle over 70 degrees is the preferred morphology after restoration [15]. Although the CO_2_ laser repair method has proven itself to be a reliable repair means, CO_2_ laser repair may also generate the re-deposition of debris, bubbles, surface deformation, and beam modulation [7]. Beam modulation originates from the slope of raised-rim a distance dependent hot spot, which threatens the downstream optical surface. Additionally, the laser output quality is also deteriorated due to modulation. Since the amount of light focused onto the hot spot is in scale with the area of the outer slope, generally, the larger the raised rim is, the higher the intensification is, to deal with the problem, Matthews [16] used a pulsed CO_2_ laser to reduce the ring-shaped rim. The Lawrence Livermore National Laboratory staff adopted a four-step “dimpling” method [17] to repair laser damage on fused silica components, which consisted of sequential operations of laser evaporation, the dimpling of a raised rim, melt re-deposition, and stress annealing. However this 4-step repair method may also bring in residual ablation or new raised-rim while repairing, to improve the problems Bass proposed an improved repair method based on a scan pattern of a set of concentric circles [17] and achieved restoration of a large damage crater with a diameter of 475 μm and a depth of 275 μm. Although scholars have conducted a lot of work on the CO_2_ laser restoration process, little research is focused on the influence of key parameters on the morphology of repaired pits; therefore, its discussion is of value.

To improve the raised rim problem, this paper combined experiment and simulation measures to explore the influence of different key restoration parameters on restoration pit morphology. A restoration strategy for migrating the raised rim problem for a large laser-induced damage crater with deep defects is then proposed and verified in an experiment. The paper is organized as follows: in Section 2, the related theoretical analysis of heat transfer and the restoration mechanism is reported, in Section 3 and Section 4, the results of the simulation and experiment are demonstrated, respectively, and in Section 5, we summarize all the work.

## 2. Restoration Mechanism Analysis

When a CO_2_ laser is irradiating the surface of fused silica, the material will absorb a lot of heat, which leads to a series of physical phenomena: temperature rise, melting, gasification and sputtering in the irradiation area [7,18,19]. 

### 2.1. Heat Transfer Parameter Configuration

The restoration process of CO_2_ laser irradiation on fused silica is a complicated non-linear process consisting of heat absorption, material phase transformation, melt flow and evaporation. Heat absorption is the main mechanism throughout the process, the morphological evolution of surface of optical components is closely related to transient temperature field. The main heat transfer parameters involved in this complicated non-linear process include laser absorptivity, environmental heat exchange coefficient, heat capacity, and heat conductivity. As a consequence of high temperatures, the coexistence of nonlinear heat transfer parameter variations, material phase transitions, and viscous flow complicate the accurate calculation of the temperature field. In that sense, to simulate temperature evolution during the process, the assumptions below are adopted:(1)Fused silica material is assumed to be isotropic during the irradiation;(2)Air convection coefficient is assumed to be constant during the irradiation;(3)Assume surface laser absorptivity of the fused silica is constant throughout the irradiation;(4)Assume melt flow is not affected by material steam recoil;

The temperature dependent heat transfer parameters of fused silica are shown in Figure 1.

### 2.2. Heat Transfer Mechanism

For CO_2_ laser irradiation on the fused silica sample depicted in Figure 2, the sample is a rectangle block, with dimension parameters a, b, h corresponding to width, length and height, respectively. The loading area of laser beam is located at the center of upper surface of sample, the radius of the beam is r.

When the laser beam is irradiating at the fused silica sample, in accordance with the law of energy conservation, the equilibrium equation is expressed as:(1)$$E_{0}=E_{a}+E_{r}+E_{p}$$
where $E_{0}$ represents the total irradiation energy, $E_{a}$ represents the energy absorption by sample, $E_{r}$ is the reflection of the laser beam, and $E_{p}$ refers to energy penetration.

As fused silica manifest a strong absorbency for a laser beam at a spectrum of 10.6 μm, assuming $E_{p}=0$ then Equation (1) can be simplified as:(2)$$E_{0}=E_{a}+E_{r}$$

As laser beam irradiates fused silica sample vertically, the laser energy absorption and energy reflection ratio are defined as [20]:(3)$$A=\frac{E_{a}}{E_{0}}=\frac{4n}{{\left(n+1\right)}^{2}+k^{2}}$$
(4)$$R=\frac{E_{r}}{E_{0}}={\left|\frac{n\prime -1}{n\prime +1}\right|}^{2}=\frac{{\left(n-1\right)}^{2}+k^{2}}{{\left(n+1\right)}^{2}+k^{2}}$$
where n′ represents the complex refractive index, obtained by
$$n^{\prime }=n-ik$$
where n represents the refractive index, and k represents the absorption coefficient, which denotes the energy decrement during propagation in absorbing medium.

For the fundamental mode Gaussian beam shown in Figure 2, according to Lambert–Beer law, the incident laser intensity is expressed as:(5)$$I(x,y,z)=\frac{P_{0}}{\pi r^{2}}e^{-2(\frac{x^{2}+y^{2}}{r^{2}})-\alpha z}$$
where $P_{0}$ is the laser power, and α is the absorption coefficient, obtained by
$$\alpha =\frac{4\pi k}{\lambda }$$
where z represents the vertical incident depth. 

Based on assumptions in Section 2.1, the governing equation concerning incident laser heating effects on the fused silica surface is
(6)$$\rho C(T)\frac{\partial T}{\partial t}=\nabla \cdot (k(T)\nabla T)+Q$$
where ρ refers to the density, t is the irradiating time, C(T) refers to the temperature dependent specific heat function, k represents the heat conductivity, and Q is the external heat load brought in by laser beam, obtained by
(7)$$Q=A\alpha \frac{P}{\pi r^{2}}exp(-2(\frac{x^{2}+y^{2}}{r^{2}})-\alpha z)$$
where P is the laser power.

Due to continuous temperature rising by laser irradiation, where the solid fused silica would turn into a molten or vapor state, assume $T_{m}$ as the phase transition temperature at time $t_{m}$, on phase boundary $f_{m}$, the temperature equals $T_{m}$. The phase boundary changes with the irradiation duration: the boundary move deeper into the solid zone as irradiation goes on, and moves towards the fused silica surface as irradiation stops. The boundary conditions are:(8)$$\left\{\begin{array}{c} T_{m}=T_{s}=T_{l}\\ -k(T)\nabla T_{s}+k(T)\nabla T_{l}=\rho L_{m}\frac{df_{m}}{dt} \end{array}\right.\;t\geq t_{m},\;f_{m}=f_{m}(t)$$
where $T_{s}$ and $T_{f}$ represent the temperature of solid and fluid phases, respectively, $L_{m}$ is the latent heat of fusion, and $t_{m}$ is the time required to reach melting point.

As the result of the temperature difference between environment and fused silica, the heat exchange between the fused silica and environment consequently occurs in restoration, and the related thermal boundaries are:(9)$$\left\{\begin{array}{c} T(x,y,z,t){|}_{t=0}=T_{0}\\ k(T)\nabla T=h(T_{0}-T) \end{array}\right.$$
where $T(x,y,z,t){|}_{t=0}$ represents the initial temperature of fused silica, h is the free convective factor and, for ambient temperature, $T_{0}=293\;K$, $h=10\;W/m^{2}\cdot K$.

### 2.3. Laser-Induced Damage Restoration Mechanism

A major factor in fused silica surface damage restoration is the combined effects of material melt and evaporation due to the rising temperature, the restoration process is shown in Figure 3a–c; when the laser is irradiating at the surface of the fused silica, the surface absorption of laser energy causes the temperature of the fused silica to rise. Then, the surface material begins to soften when it reaches the softening temperature. When the melting temperature is reached, the fused material begins to flow, forming a micro molten pool, which is slightly smaller than the laser spot. The fusion material in the molten pool flows from the micro peak of the surface to the valley, under the comprehensive action of the forces, which consist of surface tension, buoyancy and light pressure simultaneously; in the meantime, the fusion material also flows from the irradiation center to the periphery; thus, a smooth surface of fused silica is achieved during the re-solidification of the cooled material, and material in the defect area is either redistributed or removed by evaporation. In this way, the laser induced damage crater is restored. As a temperature difference always exists in a tangential direction along the molten pool surface, due to the unique laser energy distribution, the Maragoni flows occur accordingly. Since fused silica is amorphous, which means it has no fixed melting and boiling points, generally 1973 K and 2473 K are recognized as the temperature fused silica starts to melt and evaporate [7]; the corresponding isothermal surfaces represent melting interface and evaporation interface, respectively.

During the repair process, the fluid heat transfer of fused silica includes heat conduction and convection; the coupled heat transfer equation is:(10)−∇·(k(T)∇T+ρCTu)=Q
where u represents the displacement field.

For the viscous melt fluid, the velocity field and pressure field is subject to Navier–Stokes equation as follows:(11)$$-{\eta \nabla }^{2}u+\rho u\cdot \nabla u+\nabla p=F$$
where η is the dynamic viscosity, p is the pressure, and F represents the external force per unit volume. The external force consists of three different parts:

(1)Buoyancy

Buoyancy-driven flow is an internal flow of fluid caused by uneven buoyancy, which results from a density variation with the temperature field. The buoyancy $F_{float}$ is obtained by
(12)$$F_{float}=\rho (T)g$$
where ρ(T) represents the temperature dependent density, and g represents the gravity.

(2)Surface tension

Surface tension refers to the tension acting on any boundary along the liquid surface layer due to an imbalance of molecular attraction, obtained by
(13)$$\sigma_{n}=(-P_{0}+\kappa \gamma T)n$$
where γ is the derivative of the surface tension factor to the temperature, $P_{0}$ represents the atmosphere, κ refers to the curvature, and n is the normal vector of gas-liquid interface.

The temperature-dependent surface tension gradient variation also results in a Marangoni flow at the liquid–gas interface (or liquid–liquid interface), which plays an important role in occasions of crystal growth, welding, and laser or electron beam heating; the Marangoni tangential force is obtained by
(14)$$\sigma_{m}=-\gamma \cdot \nabla T\cdot l$$
where l represents the tangent vector along the interface.

(3)Light pressure

Light pressure is the result of the particle nature of light; as incident photons have a certain amount of momentum, the incident photons are either absorbed or reflected by material surface; thus, the total momentum of incident photons equals the sum of the photon momentum after the incidence and pulse on the material surface. Light pressure is obtained by [21,22]:(15)$$P_{photon}=\frac{I(1+R)}{c}$$
where I represents unitized incident energy by laser beam, and c represents the speed of light in vacuum.

## 3. Numerical Analysis

Existing research results [23] indicate that the flow field is closely linked to the temperature field. In that sense, the finite element method based on a level set is then applied to simulate the fusion and evaporation of fused silica material, so as to explore the pit morphology related to the material redistribution and removal in the restoration process. Resulting from unique Gaussian light intensity distributions and low thermal conductivity of fused silica, the irradiation center area bears most of the laser energy, and the central temperature is much higher than the lateral area. As the combined result of the temperature exponential decay viscosity, temperature dependent surface tension and Marangoni tangential force, which are the main driven forces in raised rim generation, melt material flows out of central area, quickly piles up, and eventually forms the post-restoration morphology of a raised rim with a smooth surface.

As a raised rim is the consequence of the unique temperature distribution, the synergy of the laser power, irradiation duration, and laser beam diameter have a decisive role in the formation of the temperature field during the restoration process; the separate influence of the parameters are now discussed.

(1)Laser power

In simulations, the different laser powers are set as 20 W, 30 W, 40 W, and 50 W, respectively. The laser beam diameter is 600 μm, and the total irradiation duration is 60 ms. The corresponding simulation results are shown in Figure 4.

As shown in Figure 5, a high laser power leads to a rapid increase in the raised rim height and width, and the size of raised rim increases as the power increases. That is because a high laser power leads to a high light pressure, high temperature, and surface temperature gradient, which result in a high surface tension. Thus, more material around the irradiation center flows to the near center area and piles up. 

(2)Irradiation duration

To study the effects of the irradiation duration on the raised rim dimensions, a long-term laser restoration process is observed. The power is 50 W, and laser beam diameter remains 600 μm. The total irradiation duration is set to 60 ms. 

As depicted in Figure 6 and Figure 7, through the whole duration, the width of restoration pit increases rapidly with the irradiation duration first, then gradually slowed down, so does the raised-rim height. As shown in Figure 8, the fitting line of the relationship between maximal flow speed and irradiation duration also denotes that the melt flow speed grew steadily with the irradiation duration. The phenomenon is because when the temperature keeps on rising, the heat capacity and heat conductivity of fused silica both increase accordingly, and the surface heat convection rate also increases due to a higher temperature difference; hence, as a consequence, under same laser energy input, the temperature rising rate gradually flattens, and the gradient gradually decreases to a relatively stable state with the irradiation duration, as shown in Figure 9. The changing trend eventually leads to an approximate constant surface tension, which results in a steady growth in melt flow speed. As the melt material flow only occurs in a small area around irradiation center, out of the irradiation zone the temperature field changes rather slowly, which results in a slow diameter increase in the raised rim. In terms of depth direction, the central melt flow speeds up and flows into lateral zone under a steady surface tension, which leads to a rapid restoration of pit steepening, as shown by fitting lines in Figure 10.

(3)Laser beam diameter

Laser beams with diameter of 400 μm, 600 μm, 800 μm, 1000 μm are applied; the corresponding raised rim morphologies are shown in Figure 11. The other laser parameters are a laser power of 50 W and an irradiation duration of 60 ms.

With a decrease in laser beam diameter, the power density, which was defined as the ratio of total power to irradiation area, will increase, and result in a higher maximal temperature. As the laser energy remains the same, the increase in power density will cause a smaller high-temperature area; thus, the restoration width of the laser-induced damage are confined. As shown in Figure 12, when the laser beam diameter increases, the rim width will increase correspondingly, until the maximal temperature is insufficient to evaporate the fused silica. While the repair pit depth and rim height decrease as the beam diameter increases, as shown in Figure 13 and Figure 14, the increase in laser beam diameter will result in the decrease in both the light pressure and temperature rising rate; therefore, the flow speed from the irradiation zone to the surroundings decreases as the laser diameter increases. Hence, for small and deep damage restoration, laser beams with small diameters are preferred, while for large and shallow damage craters, laser beams with large diameters are better options.

## 4. Experimental Study on Large Laser-Induced Damage Restoration

An experimental study on the restoration of a large laser-induced crater with a crack depth reaching 60 μm is then conducted. The dimension of the fused silica is 10 mm × 10 mm × 5 mm. To obtain the preferred post-restoration conical shape of a large laser-induced damage crater, the restoration method by Bass is adopted [17]. The scan pattern of one complete scan operation consists of 7 concentric circles as shown in Figure 15; the scan number of each circle is setup in a linear growth strategy as 2, 4, 6, 8, 10, 12, and 14, inward from outermost circle to the center, to fit the crater shape; the depth of the damage crater is deeper in central area, and shallower in the lateral. Since large laser spots are suitable for restoration of shallow damage craters, for large laser-induced damage craters with deep defects, a smaller laser spot is preferred; therefore, the spot diameter is set as 100 μm, the laser spot overlap ratio is 50%, and the scan speed is 0.5 mm/s, so as to heat material thoroughly.

As depicted in Figure 16, the restoration system consists of a control terminal, a water cooling machine, a laser device, and a high precision moving objective loading table. The location of the damage craters depends on the microscope camera and deflectable galvanometer.

To verify the influence of the laser energy on the restoration pit morphology, the same fused silica plane surface repair results under a laser power of 10 W, 15 W, and 20 W are compared. Through a Zygo interferometer, the restoration pit morphologies under different energies are measured; the results are shown in Figure 17.

Figure 18 depicts the corresponding outlines of the repair pits under different laser powers, the depths of the repair pit are 56.21 μm, 90.67 μm, 127.42 μm, respectively. For the width dimensions of the restoration pits, the corresponding diameters are 1.43 mm, 1.67 mm and 1.87 mm, respectively. The variation tendency is obvious: as the laser energy increases, the depth of the pit significantly deepens and steepens, as shown in Figure 13; the horizontal size also shares the same changing tendency.

The corresponding height dimensions of the raised rims are 2.64 μm, 4.63 μm, and 6.29 μm; the changing tendency indicates that a higher laser energy results in higher and wider rims. The changing tendency of the laser power to the post-restoration morphology is basically consistent with the simulation results in Section 3.

Since both high power and long laser irradiation will intensify the raised rim problem, in this case a restoration strategy is proposed: reduce the laser energy and increase the scan number simultaneously. With this strategy, the irradiation energy is dispersed, and more cooling time is offered for the melt material flow; thus, both maximal temperature and melt zone are reduced in the restoration process, and the size of the raised rim will then be restrained. Hence, a further contrast experiment between laser load cases with 10 complete scans under a laser power of 5 W, and single complete scan operation with power of 10 W, is conducted. The results are shown in Figure 19; for the load case with a laser power of 10 W, the height of the raised rim is 1.52 μm, while the height for 15 W case is 2.6 μm; the corresponding diameter for each restoration pit is 1.83 mm and 1.76 mm, respectively. In Figure 20, it can be judged from the corresponding outlines that the edge effect is significantly restrained by reducing the irradiation energy and increasing the scan number.

The photo-thermal absorption results are shown in Figure 21; for both cases, the photo-thermal absorption along raised-rim is significantly stronger than the other area For the 10 W case, the average absorption is 2.13 PPM, while the maximal value is 784.2 PPM; the numbers for the 5 W case are 1.79 PPM and 209.43 PPM, respectively. The results indicate that the proposed strategy of reducing the laser energy and increasing the scan number has lowered the absorption of the post-restoration pit, a better restoration result is achieved with the strategy.

## 5. Conclusions

To explore the influence of key laser power parameters on the post-restoration morphology in the repair process of laser-induced damage, so as to restrain the raised rim problem, different laser powers, irradiation durations and laser beam diameters were tested and compared through simulations and experiments; the conclusions below were obtained:(1)A high laser power will result in a growth in the size of a raised rim, including the rim height and width;(2)During restoration, the height and width of the raised rim increases rapidly first, then the increase rates gradually flatten, while the melt flow speed and pit depth both retain a relatively high increase rate throughout the irradiation;(3)Laser beam diameter influences the post-restoration morphology through changing power density; in small and deep damage restoration, a small diameter is preferred, while for a large and shallow damage crater, the large diameter laser beam is a preferable option;(4)By simultaneously reducing the laser power and increasing the scanning number in the patterned CO_2_ laser restoration experiment, the height of raised rim is significantly restrained.

## Figures and Tables

**Figure 1 micromachines-14-01359-f001:**
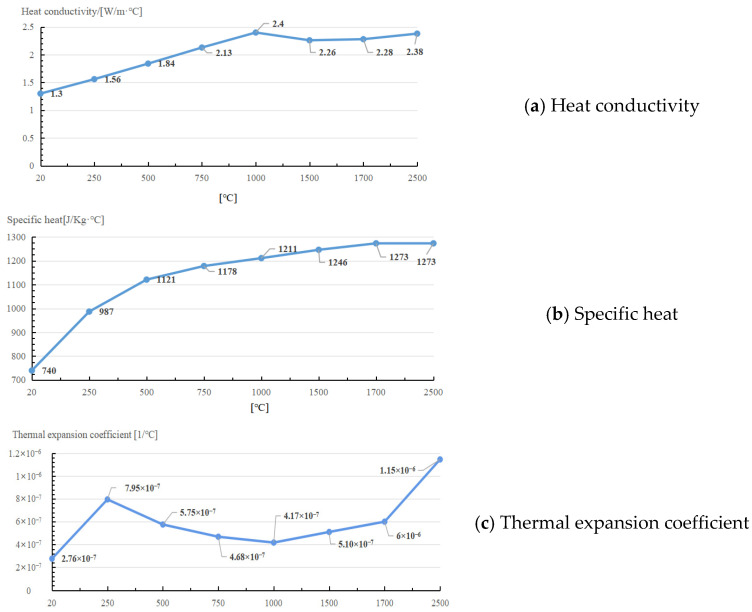
Temperature dependent heat transfer parameters of fused silica.

**Figure 2 micromachines-14-01359-f002:**
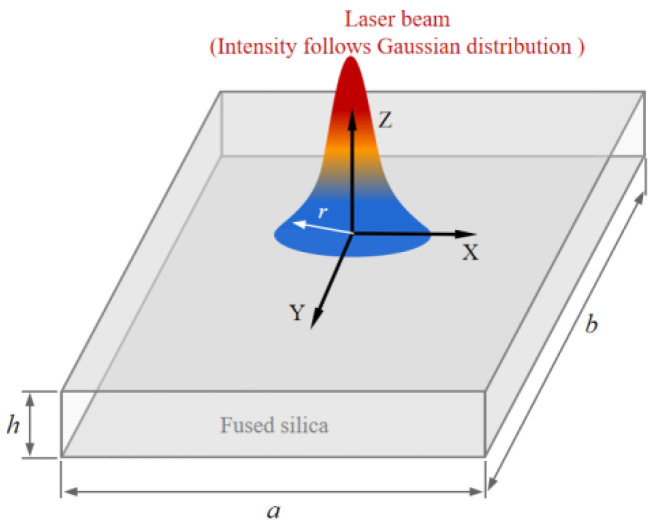
Schematic of laser irradiation on fused silica.

**Figure 3 micromachines-14-01359-f003:**
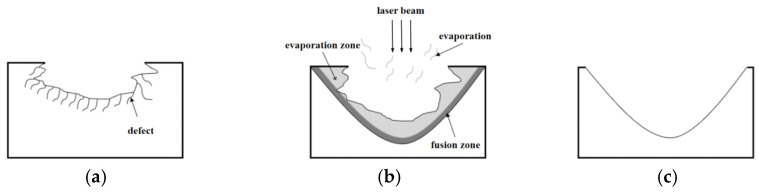
Process of laser-induced damage restoration. (**a**) Initial laser-induced damage crater. (**b**) Restoration mechanism—defective material is either evaporated or redistributed by fused flow. (**c**) Smoothened surface of post-restoration pit.

**Figure 4 micromachines-14-01359-f004:**
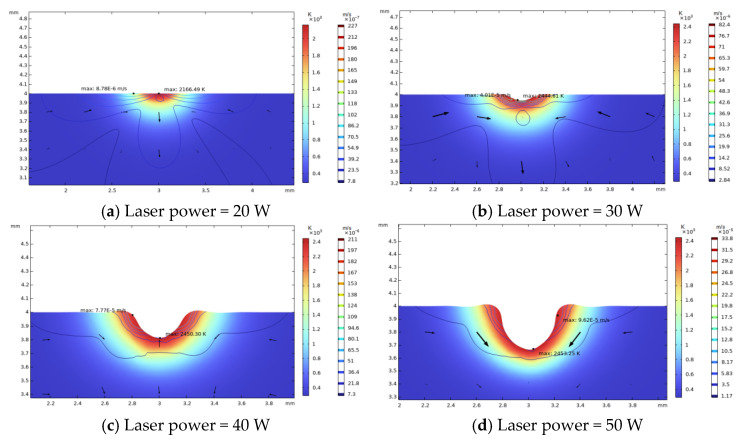
Influence of different laser powers on raised rim morphology (laser diameter 600μm, irradiation duration 60 ms).

**Figure 5 micromachines-14-01359-f005:**
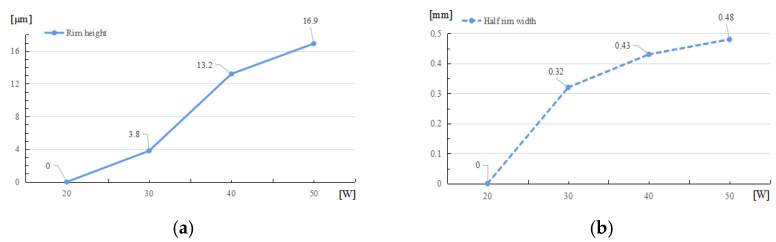
Raised rim dimensions under different laser powers. (**a**) Relationship between rim height and laser powers. (**b**) Relationship between half rim width and laser powers.

**Figure 6 micromachines-14-01359-f006:**
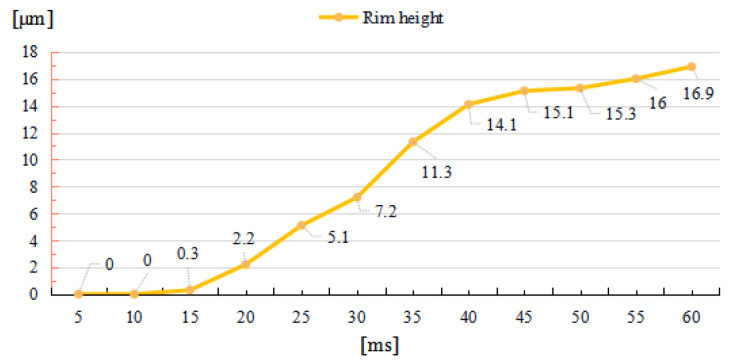
Relationship between rim height and irradiation duration.

**Figure 7 micromachines-14-01359-f007:**
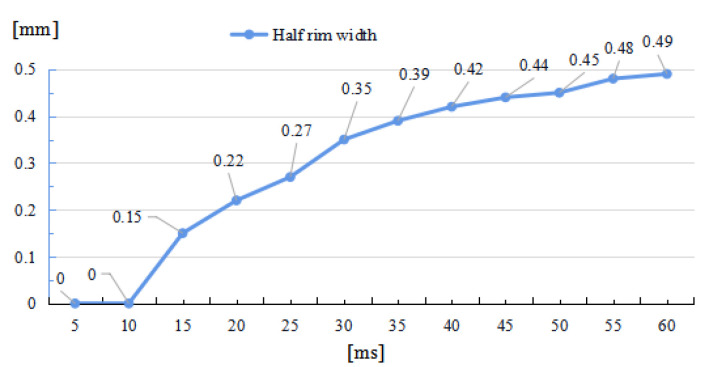
Relationship between half rim width and irradiation duration.

**Figure 8 micromachines-14-01359-f008:**
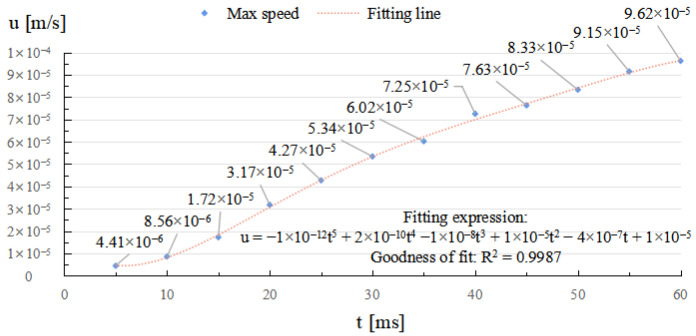
Relationship between maximal melt flow speed and irradiation duration.

**Figure 9 micromachines-14-01359-f009:**
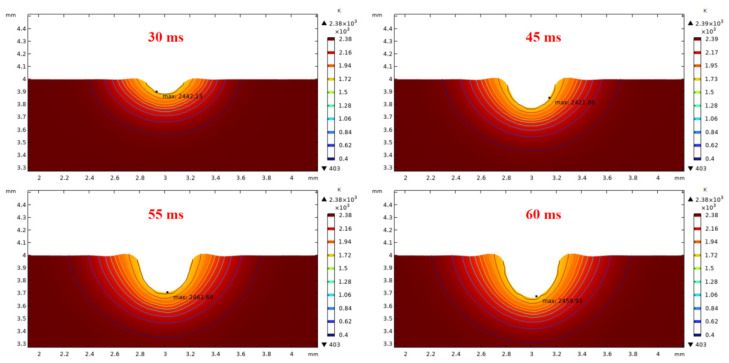
Influence of different irradiation durations on transient temperature fields and pit morphologies (laser power 50 W, laser beam diameter 600 μm).

**Figure 10 micromachines-14-01359-f010:**
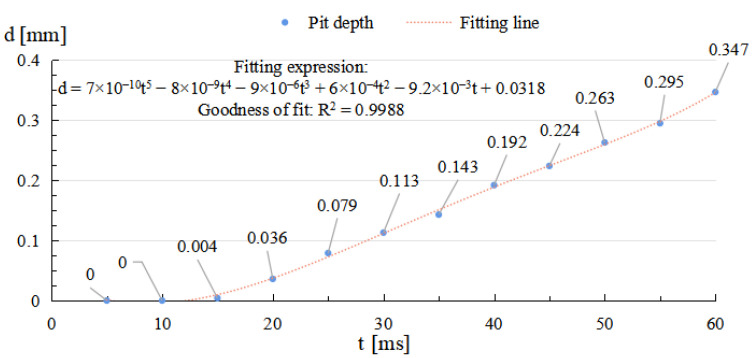
Relationship between restoration pit depth and irradiation duration.

**Figure 11 micromachines-14-01359-f011:**
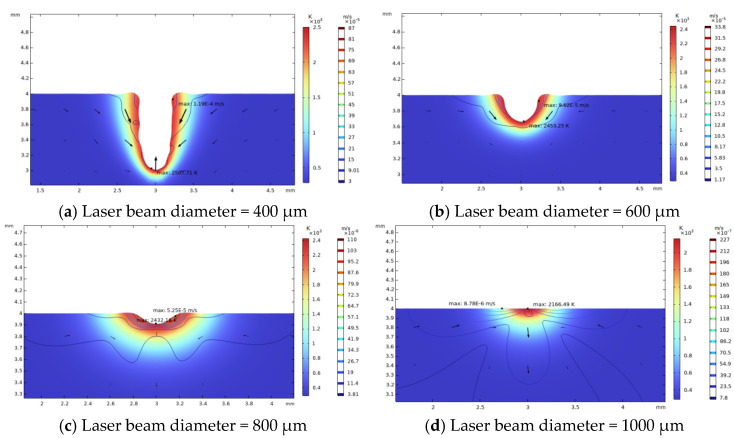
Morphology of restoration pit under different laser beam diameters (laser power 50 W, irradiation duration 60 ms).

**Figure 12 micromachines-14-01359-f012:**
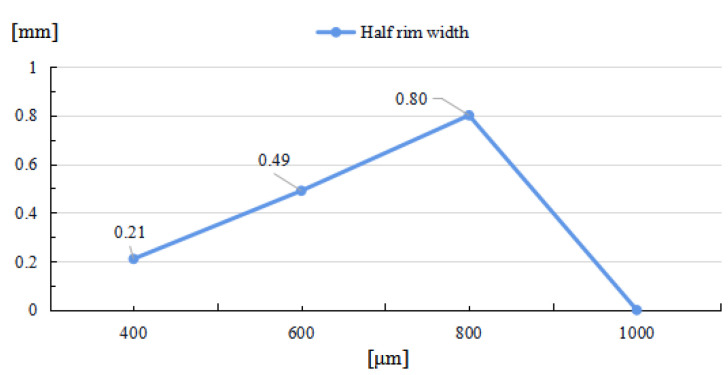
Relationship between half rim width and laser beam diameter.

**Figure 13 micromachines-14-01359-f013:**
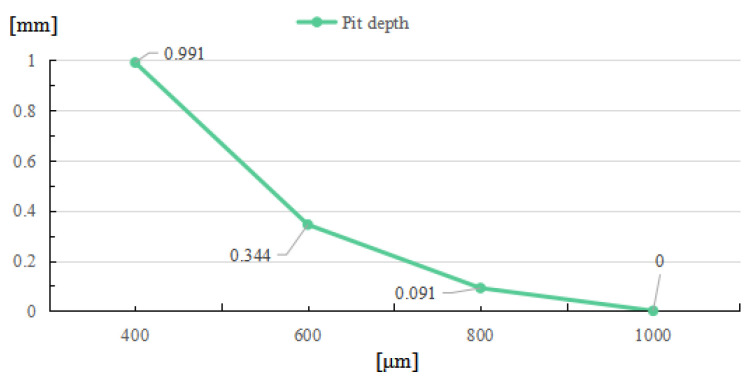
Relationship between pit depth and laser beam diameter.

**Figure 14 micromachines-14-01359-f014:**
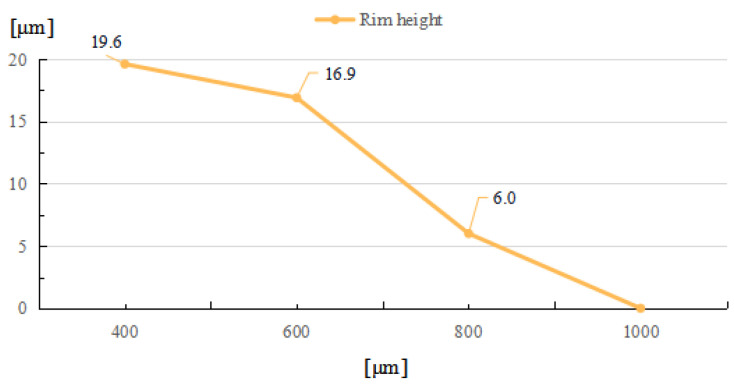
Relationship between rim height and laser beam diameter.

**Figure 15 micromachines-14-01359-f015:**
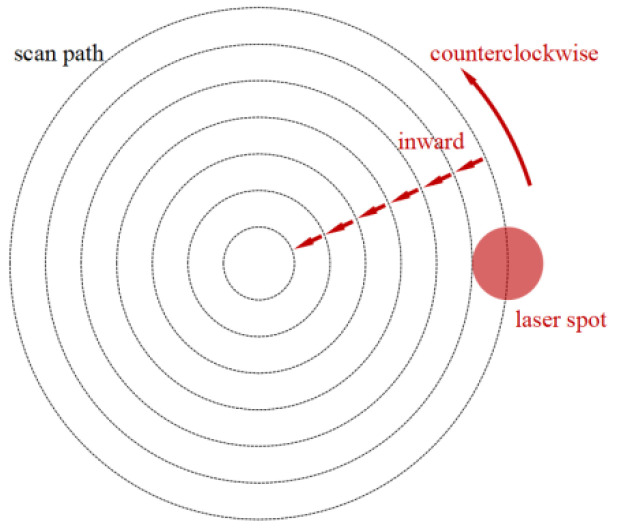
Laser scan pattern in restoration.

**Figure 16 micromachines-14-01359-f016:**
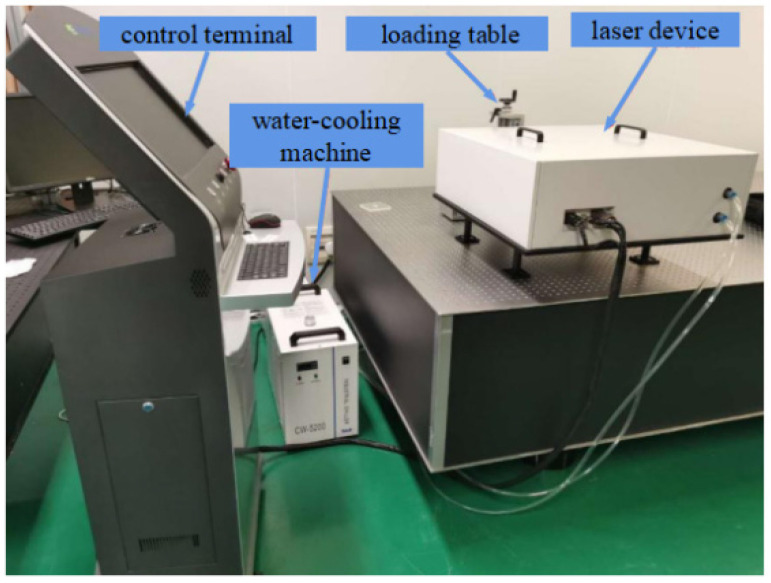
Restoration system.

**Figure 17 micromachines-14-01359-f017:**
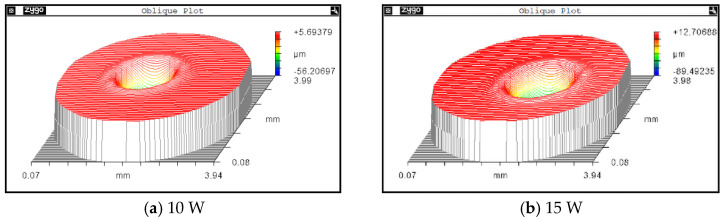
Restoration pit morphology under different laser powers.

**Figure 18 micromachines-14-01359-f018:**
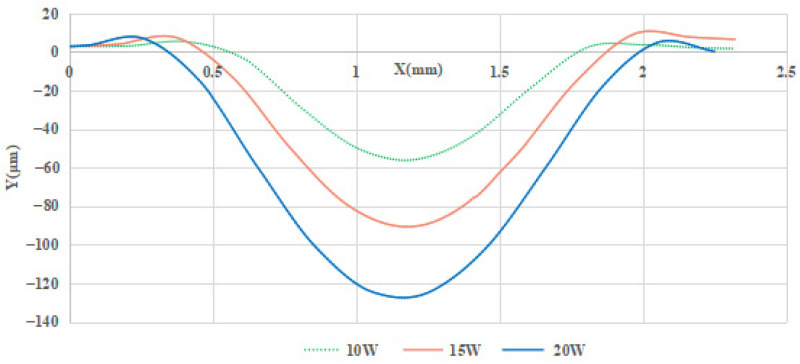
Restoration pit section contours under different laser powers.

**Figure 19 micromachines-14-01359-f019:**
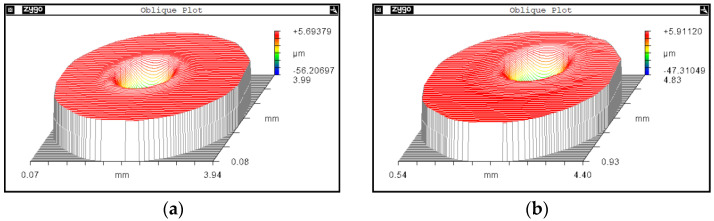
Pit morphology under different scan numbers. (**a**) Pit morphology after single complete scan. (**b**) Pit morphology after 10 complete scans.

**Figure 20 micromachines-14-01359-f020:**
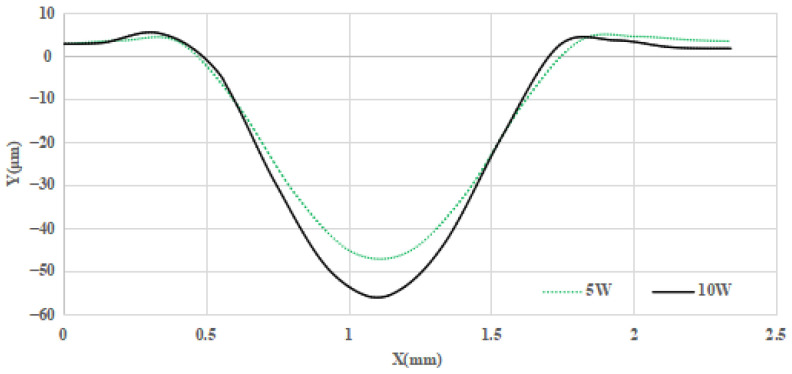
Section contours of post-restoration pit under different load cases.

**Figure 21 micromachines-14-01359-f021:**
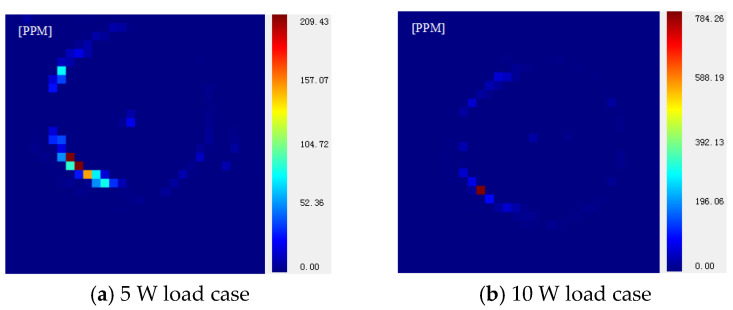
Photo-thermal absorption test results for different cases.

## Data Availability

The data presented in this study are available on request from the corresponding author. The data are not publicly available due to the data also forming part of an ongoing study.

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
