# Peer review of "Research on Laser-Induced Damage Post-Restoration Morphology of Fused Silica and Optimization of Patterned CO2 Laser Repair Strategy"

_micromachines, 2023, doi:10.3390/mi14071359_

Round 1

Reviewer 1 Report

The authors investigated the laser-fused silica interaction under different parameters such as power, time, patterning, et al. While it is important to understand quantitatively how laser can damage/restore the fused silica material as the lens system, the significance of the manuscript is generally low and does not offer enough conclusions expected for a research article.

1. It is suggested that the authors have a native English speaker to go over the whole context and make necessary revisions to the language wherever needed. Overall, we find difficulties understanding the present manuscript smoothly.

2. Some statements are ambiguous. For example, in line 39 - 41, it is stated that "The number of large aperture fused silica lenses with diameters over 430 millimeters in NIF is more than 2700 and over 2400 for LMJ." What is the unit for the values 2700 and 2400? Or is it numerical aperture, but the authors simply called it as "aperture"?

3. Please clarify the meaning of "ICF" in the abstract. Please add reference to lines 100 - 102.

4. Lines 54 - 89 performs literature review of related research works, but this part is too lengthy and needs to be shortened. Also, figure 1 is not necessary, and should be merged into other figures, if the authors would rather keep it.

5. The captions for the figures must be reworked. For example, the caption for Figure 8 reads "Speed and depth variation with irradiation time". This is by no means acceptable in a research article, since it does not provide any clear information out of the figure itself, and only causes confusion.

6. In Figure 15, it seems that the absorption is not angularly symmetrical, and the left-bottom part of the pattern has a higher portion of absorption. Why?

Taking the abstract as an example.

1. "Fused silica has become a preferred optical material in field of ICF due to its excellent performance, the laser induced damage restoration of these optical elements is of great value." This is not a grammarly correct sentence and makes readers confused.

2. "To restrain the post restoration raise rim problem in CO2 laser repair process so as to ..." We searched literatures and did not find the phrase "the post restoration raise rim problem" to be a term in the field. The authors shall at least use "post-restoration raised-rim problem" to help clarify what they are referring to, or better avoid simplifications when raising the concept for the first time.

3. Use "through" instead of "thru".

The grammar of the whole context must be carefully gone over and revised to make readers understand it successfully.

Author Response

We deeply appreciate your suggestion. Those comments are all valuable and very helpful for revising and improving our paper. We have studied comments carefully and modified the manuscript accordingly. Our point-by-point responses can be found in the attachment.

Reviewer 2 Report

Report on the manuscript: Research on laser induced damage post restoration morphology of fused silica and optimization of patterned CO2 laser repair strategy

by Xiao Shen, Ci Song, Feng Shi, Ye Tian, Guipeng Tie, Suo Qiao, Xing Peng, Wanli Zhang, Zhanqiang Hou

The manuscript reports on the influence of CO2 laser power, irradiation time and beam diameter on the laser induced damage restoration of fused silica by performing simulation and experiments. The author finds out that for a small and deeper damage restoration, a small diameter laser beam is preferred whereas for a large swallow damage restoration, a large diameter beam is preferred. The author also provide a better strategy to avoid the rim rise problem, that is instead of increasing laser power density, increase of scanning cycles reduces the rim depth and also lower photothermal absorption. The results are promising and impactful for the study of laser induced damage restoration. Therefore, I would recommend publication after considering my following comments.

Comments:

1.      The authors need to write a descriptive caption for each figure. One line caption does not provide sufficient explanatory information. Descriptive caption is beneficial to understand well the figure. For an example, the author needs to explain what left, middle and right images are in Figure 1. This is same for other figures too. In some cases, units are missing. For example, colorbar needs a unit. For an example in Figure 5, the colorbar does not have any unit.

2.      I would recommend the authors to study the influence of the pulsed laser (pulse duration time, pulse frequency, peak power) on laser induced damage restoration, at least to perform the simulation to find out if the pulsed laser behavior is beneficial than CW laser in terms of restoration.

3.      Section 2.1.: The authors mention that the restoration process is non-linear since the heat absorption is the main mechanism. However, the heat absorption is a linear process. Can the author elaborate what the non-linear process they are pointing here?

4.      Figure 6: The rim height and width increases with increase in laser power because of three reasons: high light pressure, high temperature and high surface temperature gradient. Can the authors disentangle these three effects and which of these effects do have stronger influence?

5.      Section 3, subsection 2 irradiation time: The irradiation is set to 0.06 s. However, the authors are studying the influence of irradiation time on rim height and width. Therefore, irradiation times are varying, but they are not set to a fixed value. Can the author verify this point?

6.      Figures 7 and 8: Can the authors fit those curves to extract the rate of change which would be valuable parameters in the context of discussion?

7.      Figure 15: Can the authors describe how the photothermal measurement were performed and explain why the photothermal signal is higher only at the periphery?

Author Response

(The authors gave the same response as above.)

Round 2

Reviewer 1 Report

In the revised manuscript, our comments and questions have all been properly addressed, and it can be now considered to publish.

Author Response

Dear reviewer, On behalf of my co-authors, we thank you very much for the positive and constructive comments and suggestions! It's my fault not having completely fixed the issue in last revise, terribly sorry for that.

In this turn, we check all the figures and captions, and improve all ambiguous expressions in the captions,  thank you again!